
# Application of Advection-Diffusion Equation for Nonlinearly Evolving Precipitation Field

Ji-Hoon Ha[1]

[1]AI Meteorological Research Division, National Institute of Meteorological Sciences, Jeju, 63568, Republic of Korea

*Correspondence to*: Ji-Hoon Ha (jhha223@korea.kr)

**Abstract.** Analytic solutions for the Advection-Diffusion equation have been explored in diverse scientific and engineering domains, aiming to understand transport phenomena, including heat and mass diffusion, along with the movement of water resources. Precipitation, a vital component of water resources, presents a modeling challenge due to the complex interplay between advection-diffusion effects and source terms. This study aims to improve the modeling of nonlinearly evolving

precipitation fields by specifically addressing advection-diffusion equations with time-varying source terms. Utilizing analytic solutions derived through the integral transform technique, we modeled the time-varying source term and investigated the correlation between advection-diffusion and source term effects. While the growth of the field is mainly influenced by the amplitude, size, and timescale of the source term, it can be modulated by advection and diffusion effects. When the timescale of source injection is significantly shorter than the dynamic scale of the system, advection and diffusion effects become

independent of the field growth. Conversely, when the timescale of source term injection is sufficiently long, the system evolution primarily depends on advection and diffusion effects. In turbulent regimes with strong diffusion and weak advection effects, a quasi-equilibrium state between growth and decay can be established by regulating the decay caused by advection. However, in regimes where advection effects are crucial, the decay process predominates over the growth process.

## 1 Introduction

The Advection-Diffusion equation has been widely employed to study the transport of heat, mass, and water resources in many areas of science and engineering, such as physics, atmospheric sciences, environmental sciences, and hydrology (e.g., Amiri et al. 2021; Davydova et al. 2017; Jinno et al. 1993; Kawamura et al. 1997; Kumar et al. 2010; Perez Guerrero et al. 2009; Ryu et al. 2020; Shilsar et al. 2023; van Genuchten 1981). For instance, pollution issues in the air and rivers have been studied using transport modeling based on the Advection-Diffusion equation (e.g., Amiri et al. 2021; Shilsar et al. 2023).

Additionally, atmospheric applications have been investigated, such as modeling greenhouse gases in the surface atmospheric layer (e.g., Davydova et al. 2017), and precipitation estimation and prediction (e.g., Jinno et al. 1993; Kawamura et al. 1997; Ryu et al. 2020).

Among the various applications listed above, this study particularly focuses on the application of the Advection-Diffusion equation for quantitative precipitation estimation. The precipitation field, representing a significant water resource, evolves in

a highly nonlinear manner, posing a considerable challenge in modeling. Notably, it can experience rapid development or decay due to the source term, encompassing physical processes. Although the currently proposed models cannot fully describe the complexity of precipitation evolution, such models have been extensively adopted for studying quantitative precipitation estimation and precipitation prediction (e.g., Ayzel et al. 2019; Germann & Zawadzki 2002, 2004, 2006; Kawamura et al. 1997; Lee et al. 2010; Pulkkinen et al. 2019; Ryu et al. 2020; Turner et al. 2004). For instance, using a two-dimensional

advection-diffusion model, it is demonstrated that there is a correlation between increased rainfall and turbulent diffusion, consistent with moisture injection due to strong turbulence (Kawamura et al. 1997). In addition to estimating precipitation quantitatively using observed events, weather prediction based on advection and diffusion has been extensively studied. Operational weather prediction models, based solely on advection, have been developed and adopted by meteorological





administrations (e.g., Germann & Zawadzki 2002, 2004, 2006; Lee et al. 2010; Turner et al. 2004). Indeed, many works have

also presented public libraries for nowcasting based on advection (e.g., Ayzel et al. 2019; Pulkkinen et al. 2019). Furthermore, a recent study (Ryu et al. 2020) proposed a nowcasting model that incorporates both advection and diffusion, demonstrating that the model including diffusion effects outperforms the model without diffusion. Despite the aforementioned efforts, a complete understanding of the nonlinear physics inherent in precipitation fields has not yet been achieved. While the nowcasting studies listed above ignored the contribution of the source terms, it is crucial to thoroughly examine the effects of

the source term, which includes dynamic processes such as the development and/or decay of precipitation fields.

Growth and decay patterns exhibited in the precipitation field have been examined based on the analysis using weather radar data (Atencia et al. 2017; Foresti et al. 2018; Radhakrishna et al. 2012; Tang & Matyas 2018). For instance, to analyze the observed data, the source term has often been considered as zero-mean Gaussian noise in time and space (Jinno et al. 1993; Kawamura et al. 1997). This assumption is based on the temporal variation of weather conditions, and the solution remains

continuous over the time domain. Through error analysis using MAPLE prediction, Atencia et al. 2017 demonstrated that the growth and decay in the diurnal cycle are significant for accurate precipitation nowcasting. Along with such quasi-periodic variations, the rapid growth and decay of fields have also been exhibited typically in precipitation fields, which cannot be accurately modeled using the quasi-periodic variation model. For instance, the orographic effect could contribute to the growth and decay of precipitation fields (Foresti et al. 2018). Additionally, the source terms associated with deep convective clouds

and the amount of vapor on such cloud tops (e.g., So and Shin 2018) and moisture injection by jets (e.g., Lee et al. 2008) are also prominent for the growth of precipitation intensity.

Due to the challenges in understanding the nonlinear evolutionary patterns of precipitation, models based on Deep Learning methods, which are free from relying on physics, have recently been developed by training data from weather radar, satellites, and surface observations (e.g., Shi et al. 2015, 2017; Agrawal et al. 2019; Ayzel et al. 2020; Choi et al. 2021; Ha & Lee 2023a;

Ha & Lee 2023b; Kim & Hong 2022; Ko et al. 2022; Ravuri et al. 2021; Sønderby et al. 2020). The forecasting performance of these data-driven models is better than that of models based on advection and diffusion. However, the results obtained from Deep Learning models are difficult to explain due to the absence of physics.

In this work, we examined the effects of the time-varying source term in the Advection-Diffusion equation and its applications on quantitative precipitation estimation and prediction. We first adopted the generalized integral transform technique (e.g.,

Perez Guerrero et al. 2009) to obtain the analytic solutions of the Advection-Diffusion equation with periodic boundary conditions. We then analyzed the behavior of the analytic solutions in the turbulent system and the intermediate system where both advection and diffusion effects are significant. Notably, the effects of the source term on the field growth and the effects of advection-diffusion on the field decay were mainly studied.

The paper is organized as follows: In section 2, we describe the conditions for solving the equation. Then, in section 3, we

present the analytic solution based on the integral transform technique. In section 4, we provide case studies and numerical results. Finally, in section 5, we offer a summary.

## 2 Basic assumptions

For an incompressible fluid system, we consider the boundary value problem for an Advection-Diffusion equation:

$$\frac{\partial R(x,y,z,t)}{\partial t} + \left(u\frac{\partial}{\partial x} + v\frac{\partial}{\partial y} + w\frac{\partial}{\partial z}\right)R(x,y,z,t) - \left(D_x\frac{\partial^2}{\partial x^2} + D_y\frac{\partial^2}{\partial y^2} + D_z\frac{\partial^2}{\partial z^2}\right)R(x,y,z,t) = S(x,y,z,t). \quad (1)$$

Here, $R(x,y,z,t)$ represents the field undergoing advection and diffusion, $u, v, w$ denote the velocity components of advection. $D_x, D_y, D_z$ are the diffusion coefficients, and $S(x,y,z,t)$ represents the source term including physical processes.

The variables of Equation (1), $x, y, z, t$, satisfy the following conditions:

$$0 \leq t \leq T_{sys}; \; 0 \leq x \leq L_{sys}; \; 0 \leq y \leq L_{sys}; \; 0 \leq z \leq L_{sys}, \quad (2)$$





where $T_{sys}$, $L_{sys}$ represent the dynamical temporal and spatial scales of the physical system, respectively. We then define the

dynamical evolution speed as $V_{sys} \equiv L_{sys}/T_{sys}$. $V_{sys}$ is the upper limit of the advection speed, $V_{adv} = (u, v, w)$. In this work, $V_{adv}$ is assumed to be a stationary field. Indeed, it has been demonstrated that the relative importance is in the evolution of advection fields, and it has been concluded that uncertainties arising from the advection fields are less significant than uncertainties in the evolution of the precipitation field (Bowler et al. 2006).

By assuming isotropic diffusion ($D \approx D_x \approx D_y \approx D_z$), we define the characteristic spatial and temporal scales as follows:

$L_{diff}$, $T_{diff}$,

$$L_{diff} = \frac{2D}{V_{adv}}; \; T_{diff} = \frac{4D}{V_{adv}^2} = \frac{2L_{diff}}{V_{adv}}. \, (3)$$

According to the conditions of the physical medium, the precipitation field can be categorized into three different regimes: the turbulent regime (i.e., diffusion-dominant regime), the intermediate regime, and the advection-dominant regime. These regimes can be classified based on the characteristic scales, $V_{adv}, L_{diff}, T_{diff}$.

$$\text{turbulent regime: } \frac{V_{adv}}{V_{sys}} \ll 1; \frac{L_{diff}}{L_{sys}} \gg 1; \frac{T_{diff}}{T_{sys}} \gg 10, \, (4)$$

$$\text{intermediate regime: } \frac{V_{adv}}{V_{sys}} \sim 0.1 - 1; \frac{L_{diff}}{L_{sys}} \sim 0.1 - 1; \frac{T_{diff}}{T_{sys}} \sim 0.1 - 10, \, (5)$$

$$\text{advection dominant regime: } \frac{V_{adv}}{V_{sys}} \sim 1; \frac{L_{diff}}{L_{sys}} \ll 1; \frac{T_{diff}}{T_{sys}} \ll 1. \, (6)$$

To investigate the system including both the advection and diffusion effects, this work mainly considers the turbulent and intermediate regimes.

**3 Analytic solutions of Advection-Diffusion equation**

The analytic solution of the Advection-Diffusion equation has been extensively studied using the Integral Transform technique (e.g., Amiri et al. 2021; Perez Guerrero et al. 2009; Kumar et al. 2014; Shilsar et al. 2023), including the Generalized Integral Transform Technique (GITT; Cotta 1993; Perez Guerrero et al. 2009).

The GITT method can be summarized as follows: (1) Defining the unknown function (i.e., the analytic solution) as a series

expansion of eigenfunctions, (2) Converting the partial differential equation into ordinary differential equations based on integral and inverse transforms, and (3) Solving the ordinary differential equations and deriving the unknown functions using the inverse transform.

In this work, we employ the GITT method, which has been proposed in previous studies (see Perez Guerrero et al. 2009; Kumar et al. 2010), to derive the analytic solution of Equation (1). By employing a change of variables, the solution,

$R(x, y, z, t)$ can be expressed as follows:

$$R(x, y, z, t) = \theta(x, y, z, t) \exp\big(a_1 x + a_2 y + a_3 z + \, t(b_1 + b_2 + b_3)\big). \, (7)$$

Here, $\theta(x, y, z, t)$ represents the new function describing the intensity of the field, while the exponential term is the term corresponding to a traveling wave with arbitrary parameters, $a_1, a_2, a_3, b_1, b_2$ and $b_3$ (physically, these parameters represent wavenumbers and frequencies). By using Equation (7), Equation (1) can be rewritten as follows:






$$\frac{\partial \theta(x,y,z,t)}{\partial t} + \theta(x,y,z,t)\big[(b_1 + b_2 + b_3) + a_1(-D_x a_1 + u) + a_2(-D_y a_2 + v) + a_3(-D_z a_3 + w)\big]$$

$$+ (-2D_x a_1 + u)\frac{\partial \theta(x,y,z,t)}{\partial x} - D_x \frac{\partial^2 \theta(x,y,z,t)}{\partial x^2}$$

$$+ (-2D_y a_2 + v)\frac{\partial \theta(x,y,z,t)}{\partial y} - D_y \frac{\partial^2 \theta(x,y,z,t)}{\partial y^2}$$

$$+ (-2D_z a_3 + w)\frac{\partial \theta(x,y,z,t)}{\partial z} - D_z \frac{\partial^2 \theta(x,y,z,t)}{\partial z^2}$$

$$= \frac{S(x,y,z,t)}{\exp\big(a_1 x + a_2 y + a_3 y + t(b_1 + b_2 + b_3)\big)}. \,(8)$$

In accordance with the physical context, we choose the parameters $a_1, a_2, a_3, b_1, b_2$ and $b_3$ as follows:

$$a_1 = \frac{u}{2D_x}; \; a_2 = \frac{v}{2D_y}; \; a_3 = \frac{w}{2D_z}; b_1 = -\frac{u^2}{4D_x}; \; b_2 = -\frac{v^2}{4D_y}; \; b_3 = -\frac{w^2}{4D_z}. \,(9)$$

The $a_1, a_2, a_3$ indicate the wavenumber of diffusion, while $b_1, b_2, b_3$ denote the frequency of diffusion. With these parameters, we obtain the following equation,

$$\frac{\partial \theta(x,y,z,t)}{\partial t} - D_x \frac{\partial^2 \theta(x,y,z,t)}{\partial x^2} - D_y \frac{\partial^2 \theta(x,y,z,t)}{\partial y^2} - D_z \frac{\partial^2 \theta(x,y,z,t)}{\partial z^2}$$


$$= \frac{S(x,y,t)}{\exp\left(\frac{u}{2D_x}x + \frac{v}{2D_y}y + \frac{w}{2D_z}z - t\left(\frac{u^2}{4D_x} + \frac{v^2}{4D_y} + \frac{w^2}{4D_z}\right)\right)}, \,(10)$$

with the initial condition, $\theta(x,y,z,t=0)$,

$$\theta(x,y,z,t=0) = \frac{R(x,y,z,t=0)}{\exp\left(\frac{u}{2D_x}x + \frac{v}{2D_y}y + \frac{w}{2D_z}z\right)}. \,(11)$$

To apply the GITT method, we formulate the eigenvalue problem by solving the same boundary conditions as those of $\theta(x,y,z,t)$. Then, the problem is defined as solving the equation for the nontrivial solutions, $\psi(x,y,z)$,


$$\nabla^2 \psi(x,y,z) + \mu^2 \psi(x,y,z) = 0. \,(12)$$

Here, the nontrivial solutions, $\psi(x,y,z) \equiv \psi_i(x,y,z)$, satisfy orthogonality,

$$\int_V \psi_i(x,y,z)\psi_j(x,y,z)\, d\bar{V} = N_i \delta_{ij}, \,(13)$$

where $N_i$ is the normalization factor and $\delta_{ij}$ is the Kronecker delta. Using this orthogonality property, the transform pair (i.e., forward, Equation (14) and inverse, Equation (15)) can be derived as:


$$\bar{\theta}_i(t) = \int_V \Psi_i(x,y,z)\theta(x,y,z,t)\, d\bar{V}, \,(14)$$

$$\theta(x,y,z,t) = \sum_{i=1}^{\infty} \Psi_i(x,y,z)\bar{\theta}_i(t), \,(15)$$

where $\Psi_i(x,y,z) \equiv \psi_i(x,y,z)/\sqrt{N_i}$ is the normalized eigenfunctions. By implementing the operator $\int_V \Psi_i(x,y,z)\, d\bar{V}$, to Equation (12) with Equations. (14) and (15), the integral transformation of Equation (12) becomes a set of ordinary differential equations:


$$\frac{d\bar{\theta}_i(t)}{dt} + \mu_i^2 \bar{\theta}_i(t) = \bar{S}_i(t); i = 1,2,\dots. \,(16)$$

Here, $\bar{S}_i(t)$ represents the volume-integrated source-term defined as,

$$\bar{S}_i(t) = \int_V \Psi_i(x,y,z)\frac{S(x,y,z,t)}{\exp\left(\frac{u}{2D_x}x + \frac{v}{2D_y}y + \frac{w}{2D_z}z - t\left(\frac{u^2}{4D_x} + \frac{v^2}{4D_y} + \frac{w^2}{4D_z}\right)\right)}\, d\bar{V}, \,(17)$$




and the initial conditions in Equation (11) can be transformed as follows:

$$\bar{\theta}_i(t=0) = \int_V \Psi_i(x,y,z) \frac{R(x,y,z,t=0)}{\exp\left(\frac{u}{2D_x}x + \frac{v}{2D_y}y + \frac{w}{2D_z}z\right)} d\bar{V}. \quad (18)$$

Using Equations (17) and (18), Equation (16) can be solved with the follow analytic solution:

$$\bar{\theta}_i(t) = \exp(-\mu_i^2 t)\left[\bar{\theta}_i(t=0) + \int_0^t \bar{S}_i(\tau)\exp(\mu_i^2 \tau)d\tau\right]. \quad (19)$$

Finally, we obtain $\theta(x,y,t)$ and $R(x,y,t)$ as follows:

$$\theta(x,y,z,t) = \sum_{i=1}^{\infty} \Psi_i(x,y,z)\exp(-\mu_i^2 t)\left[\bar{\theta}_i(t=0) + \int_0^t \bar{S}_i(\tau)\exp(\mu_i^2 \tau)d\tau\right], \quad (20)$$

$$R(x,y,z,t) = \theta(x,y,z,t)\exp\left(\frac{u}{2D_x}x + \frac{v}{2D_y}y + \frac{w}{2D_z} - t\left(\frac{u^2}{4D_x} + \frac{v^2}{4D_y} + \frac{w^2}{4D_z}\right)\right). \quad (21)$$

The exponential term in Equation (21) represents the traveling wave, and $\theta(x,y,z,t)$ describes the time evolution of field
intensity. In particular, the term, $\exp(-\mu_i^2 t)$ in $\theta(x,y,z,t)$ indicates the field decay due to advection-diffusion. Such effects
could regulate the growth of field intensity, and here we examine the effects of advection-diffusion decay in the presence of
an arbitrary source-term, $\bar{S}_i(\tau) > 0$. Based on the relative importance of $\bar{\theta}_i(t=0)$ and $\int_0^t \bar{S}_i(\tau)\exp(\mu_i^2 \tau)d\tau$ in Equation (20),
we consider three different evolution patterns summarized as follows:

(a)  Evolution dominated by decay: purely decaying field due to the Advection and Diffusion:

$$\bar{\theta}_i(t=0) \gg \int_0^t \bar{S}_i(\tau)\exp(\mu_i^2 \tau)d\tau, \quad (22)$$

(b)  Evolution, including both decay and growth: The effects of field decay due to the Advection and Diffusion are com
parable to the growth effects driven by the source-term:

$$\bar{\theta}_i(t=0) \sim \int_0^t \bar{S}_i(\tau)\exp(\mu_i^2 \tau)d\tau, \quad (23)$$

(c)  Evolution dominated by growth: purely growing field due to the source-term:

$$\bar{\theta}_i(t=0) \ll \int_0^t \bar{S}_i(\tau)\exp(\mu_i^2 \tau)d\tau. \quad (24)$$

In the following section, we discuss the applications of the analytic solutions with various types of source-terms.

### 4 Applications of Advection-Diffusion equation

In this section, we examine the physical implications of the analytic solution described in Section 3 by considering various
applicable cases. We also report the results of numerical experiments. For the sake of simplicity, we here consider the solution
of a one-dimensional system.

### 4.1 One-dimensional solution and parameterization of advection and diffusion effects

Hereafter, we express all variables as a dimensionless form:

$$\tilde{x} = \frac{x}{L_{sys}}; \; \tilde{t} = \frac{t}{T_{sys}}. \quad (25)$$

The analytic solution of the one-dimensional equation is then summarized as follows:


$$R(\tilde{x}, \tilde{t}) = \exp\left(\tilde{L}_{diff,x}^{-1}\tilde{x} - \tilde{T}_{diff,x}^{-1}\tilde{t}\right) \times \sum_{i=1}^{\infty} \Psi_i(\tilde{x}) \exp\left(-\mu_i^2 T_{sys}\tilde{t}\right)\left[\bar{\theta}_i(\tilde{t}=0) + \int_0^{\tilde{t}} \bar{S}_i(\tilde{\tau})\exp\left(\mu_i^2 T_{sys}\tilde{\tau}\right)d\tilde{\tau}\right], (26)$$

where $\tilde{L}_{diff,x} = 2D_x/uL_{sys}$, and $\tilde{T}_{diff,x} = 4D_x/u^2 T_{sys} = 2\tilde{L}_{diff,x}(u/V_{sys})^{-1}$ are the dimensionless diffusion length-scale

and time-scale, respectively.

We here introduce a set of eigenvalues to specify the eigenfunctions for the eigenvalue problem:

$$\beta_i = i\pi; \ \mu_i = \frac{\beta_i\sqrt{D_x}}{L_{sys}}; i = 1,2,\dots (27)$$

The norms and the normalized eigenfunctions are then written as

$$N_i = \frac{1}{2}; \ \Psi_i(\tilde{x}) = \sqrt{2}\sin(\beta_i\tilde{x}). (28)$$

Note that these eigenfunctions and eigenvalues satisfy the following orthogonality property:

$$\int_0^1 \Psi_i(\tilde{x})\Psi_j(\tilde{x})d\tilde{x} = \delta_{ij}. (29)$$

With this form of eigenfunctions and eigenvalues, the quantity $\left(\mu_i^2 T_{sys}\right)^{-1}$ indicates the decay timescale of each eigenfunctions,

$$\left(\mu_i^2 T_{sys}\right)^{-1} = \beta_i^{-2}\frac{L_{sys}^2}{D_x}T_{sys}^{-1} = 2\beta_i^{-2}\tilde{L}_{diff,sys}^{-1}, (30)$$

where $\tilde{L}_{diff,sys} \equiv 2D_x/(V_{sys}L_{sys})$ is the systematic diffusion length. While the diffusion timescale, $\tilde{T}_{diff,x}$ becomes larger

when the diffusion effect is dominant (with a large $\tilde{L}_{diff,x}$), $\left(\mu_i^2 T_{sys}\right)^{-1}$ is independent of such parameters, but it only depends

on the systematic diffusion length, $\tilde{L}_{diff,sys}$.

In the analytic solutions described by equation (26), the only two free parameters, $\tilde{L}_{diff,x}$, and $u/V_{sys}$ control all effects driven

by advection and diffusion. In the following sections, we parametrized the advection and diffusion effects using the parameters,

$u/V_{sys}$ and $\tilde{L}_{diff,x}$ to address the turbulent and intermediate regimes (i.e., equations (4) & (5)). For the turbulent regimes, we

adopted the parameters, $\left(\tilde{L}_{diff,x}^{-1}, u/V_{sys}\right) = (0.1, 0.01), (0.5, 0.05)$. For the intermediate regimes, the parameters with higher

advection speed were used: $(\tilde{L}_{diff,x}^{-1}, u/V_{sys}) = (1, 0.1), (2.5, 0.25), (5, 0.5)$.

### 4.2 Evolution dominated by decay

From equation (26), the source-free solution with $\bar{S}_i(\tilde{\tau}) \approx 0$(i.e., $S(\tilde{x}, \tilde{t}) \approx 0$) is simply described as:

$$R(\tilde{x}, \tilde{t}) \approx \exp\left(\tilde{L}_{diff,x}^{-1}\tilde{x} - \tilde{T}_{diff,x}^{-1}\tilde{t}\right)\sum_{i=1}^{\infty} \bar{\theta}_i(\tilde{t}=0)\Psi_i(\tilde{x})\exp\left(-\mu_i^2 T_{sys}\tilde{t}\right). (31)$$

Boundary conditions are given as follows:

$$R(\tilde{x}, \tilde{t}=0) = 1; 0 \le \tilde{x} \le 1, (32)$$

$$R(0, \tilde{t}) = 0; \ R(1, \tilde{t}) = 0; \tilde{t} > 0. \ (33)$$

From the boundary conditions, $\bar{\theta}_i(\tilde{t}=0)$ becomes:

$$\bar{\theta}_i(t=0) = \int_0^1 \Psi_i(\tilde{x})R(\tilde{x}, \tilde{t}=0)\exp\left(-\tilde{L}_{diff,x}^{-1}\tilde{x}\right)d\tilde{x}$$

$$= \int_0^1 \Psi_i(\tilde{x})\exp\left(-\tilde{L}_{diff,x}^{-1}\tilde{x}\right)d\tilde{x}$$

$$= \sqrt{2}\frac{\exp\left(-\tilde{L}_{diff,x}^{-1}\right)\left[\beta_i\exp\left(\tilde{L}_{diff,x}^{-1}\right) - \tilde{L}_{diff,x}^{-1}\sin\beta_i - \beta_i\cos\beta_i\right]}{\tilde{L}_{diff,x}^{-2} + \beta_i^2}. \ (34)$$

In the turbulent regime, $\tilde{L}_{diff,x}^{-1} \ll 1$, for instance, the $\bar{\theta}_i(t=0)$ can be approximated as




$$\bar{\theta}_i(t=0) \approx \frac{\sqrt{2}(1-\cos\beta_i)}{\beta_i} = \begin{cases} 2\sqrt{2}/\beta_i, & for\ odd\ i \\ 0, & for\ even\ i \end{cases}. (35)$$

This indicates the evolution of system could be insensitive to the diffusion effects for sufficiently larger diffusion length. When

focusing solely on the time-dependent term, we can summarize the solutions as follows:

$$R(\tilde{x},\tilde{t}) \propto \sum_{i=1}^{\infty} \exp\left(-\left(\tilde{T}_{diff,x}^{-1} + \mu_i^2 T_{sys}\right)\tilde{t}\right), (36)$$

$$\frac{dR(\tilde{x},\tilde{t})}{d\tilde{t}} \propto -\sum_{i=1}^{\infty} \left(\tilde{T}_{diff,x}^{-1} + \mu_i^2 T_{sys}\right) \exp\left(-\left(\tilde{T}_{diff,x}^{-1} + \mu_i^2 T_{sys}\right)\tilde{t}\right). (37)$$

Without the source term, $R(\tilde{x},\tilde{t})$ decays as time, $\tilde{t}$, increases and the negative sign of $dR(\tilde{x},\tilde{t})/d\tilde{t}$ is obvious for the decaying

field.


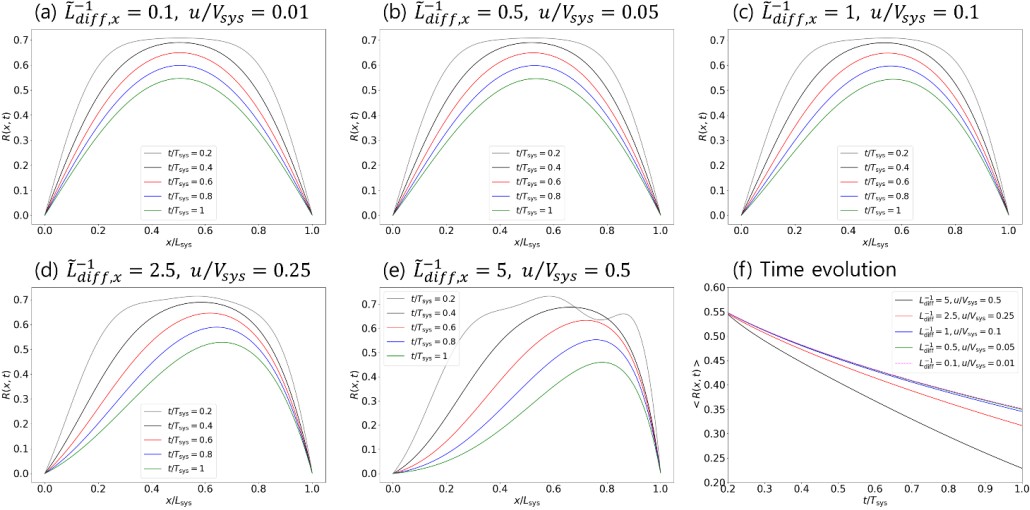

**Figure 1: (a) – (e): Time evolution of field, $R(\tilde{x},\tilde{t})$ with a set of parameters, $\left(\tilde{L}_{diff,x}^{-1}, u/V_{sys}\right) = (0.1, 0.01), (0.5, 0.05), (1, 0.1), (2.5, 0.25), (5, 0.5)$. (f) Time evolution of the spatially averaged field, $\langle R(\tilde{x},\tilde{t}) \rangle$.**

To investigate the parameter dependence of the analytic solutions, we performed numerical experiments with the following

set of parameters: $\left(\tilde{L}_{diff,x}^{-1}, u/V_{sys}\right) = (0.1, 0.01), (0.5, 0.05), (1, 0.1), (2.5, 0.25), (5, 0.5)$. Panels (a)-(e) of Figure 1 show the

time evolution of $R(\tilde{x},\tilde{t})$ with five different parameter sets. In the turbulent regime with $\tilde{L}_{diff,x}^{-1} \leq 1$ (panels (a) – (c)), the time

evolution of $R(\tilde{x},\tilde{t})$ weakly depends on the diffusion effects. In the intermediate regime with $\tilde{L}_{diff,x}^{-1} \geq 1$ (panels (c) – (e)),

the field decay becomes more prominent with stronger advection effects (or, weaker diffusion effects). The time evolution of

spatially averaged fields, $\langle R(\tilde{x},\tilde{t}) \rangle$ is shown in panel (f) to compare the spatially averaged decay rate, $d\langle R(\tilde{x},\tilde{t}) \rangle/d\tilde{t}$. Notably,

although the advection speed increases 2 times, the decay rate only increases 1.36 times since the diffusion effects regulate the

decay effects.





### 4.3 Evolution with stationary source term

In the following sections, we take into account the non-zero source term in the analytical solution, as described in Equation (26). We begin by considering a straightforward scenario in which the source term is stationary, represented as $S(\tilde{x}, \tilde{t}) = constant$. Such an assumption characterizes the presence of constant injection into the system, making it suitable for the analysis of persistent field lasting for a significant timescale. In the case where $S(\tilde{x}, \tilde{t}) \approx 1$, the expression for $\bar{S}_i(\tilde{\tau})$ is derived as follows:

$$\bar{S}_i(\tilde{\tau}) = \int_0^1 \Psi_i(\tilde{x}) S(\tilde{x}, \tilde{t}) \exp\left(-\tilde{L}_{diff,x}^{-1}\tilde{x} + \tilde{T}_{diff,x}^{-1}\tilde{\tau}\right) d\tilde{x}$$

$$= \sqrt{2} \frac{\exp\left(\tilde{T}_{diff,x}^{-1}\tilde{\tau} - \tilde{L}_{diff,x}^{-1}\right)\left[\beta_i \exp\left(\tilde{L}_{diff,x}^{-1}\right) - \tilde{L}_{diff,x}^{-1}\sin\beta_i - \beta_i\cos\beta_i\right]}{\tilde{L}_{diff,x}^{-2} + \beta_i^2}$$

$$\propto \exp\left(\tilde{T}_{diff,x}^{-1}\tilde{\tau}\right). (38)$$

Then the integral, $I_{s,i}(\tilde{t})$ including $\bar{S}_i(\tilde{\tau})$ becomes:

$$I_{s,i}(\tilde{t}) = \int_0^{\tilde{t}} \bar{S}_i(\tilde{\tau}) \exp\left(\mu_i^2 T_{sys}\tilde{\tau}\right) d\tilde{\tau}$$

$$= \sqrt{2} \frac{\exp\left(-\tilde{L}_{diff,x}^{-1}\right)\left[\exp\left(\tilde{t}\left(\tilde{T}_{diff,x}^{-1} + \mu_i^2 T_{sys}\right)\right) - 1\right]\left[\beta_i \exp\left(\tilde{L}_{diff,x}^{-1}\right) - \tilde{L}_{diff,x}^{-1}\sin\beta_i - \beta_i\cos\beta_i\right]}{\left(\tilde{L}_{diff,x}^{-2} + \beta_i^2\right)\left(\tilde{T}_{diff,x}^{-1} + \mu_i^2 T_{sys}\right)}. (39)$$

In the turbulent regime, $\tilde{L}_{diff,x}^{-1} \ll 1$, $I_{s,i}(\tilde{t})$ becomes:

$$I_{s,i}(\tilde{t}) \approx \frac{\sqrt{2}(1 - \cos\beta_i)}{\beta_i} \frac{\left[\exp\left(\tilde{t}\left(\tilde{T}_{diff,x}^{-1} + \mu_i^2 T_{sys}\right)\right) - 1\right]}{\left(\tilde{T}_{diff,x}^{-1} + \mu_i^2 T_{sys}\right)}, (40)$$

which demonstrates that the growth due to the source term also could be independent of the diffusion effects. When focusing solely on the time-dependent term, we can summarize the results as follows:

$$I_{s,i}(\tilde{t}) \propto \frac{1}{\tilde{T}_{diff,x}^{-1} + \mu_i^2 T_{sys}}\left[\exp\left(\left(\tilde{T}_{diff,x}^{-1} + \mu_i^2 T_{sys}\right)\tilde{t}\right) - 1\right], (41)$$

$$R(\tilde{x}, \tilde{t}) \propto \sum_{i=1}^{\infty} \exp\left(-\left(\tilde{T}_{diff,x}^{-1} + \mu_i^2 T_{sys}\right)\tilde{t}\right)\left[\bar{\theta}_i(\tilde{t} = 0) + I_{s,i}(\tilde{t})\right]$$

$$= \sum_{i=1}^{\infty} \exp\left(-\left(\tilde{T}_{diff,x}^{-1} + \mu_i^2 T_{sys}\right)\tilde{t}\right)\left\{\bar{\theta}_i(t = 0) + \frac{1}{\tilde{T}_{diff,x}^{-1} + \mu_i^2 T_{sys}}\left[\exp\left(\left(\tilde{T}_{diff,x}^{-1} + \mu_i^2 T_{sys}\right)\tilde{t}\right) - 1\right]\right\}, (42)$$

$$\frac{dR(\tilde{x}, \tilde{t})}{d\tilde{t}} \propto \sum_{i=1}^{\infty} \exp\left(-\left(\tilde{T}_{diff,x}^{-1} + \mu_i^2 T_{sys}\right)\tilde{t}\right). (43)$$

According to Equations (42) and (43), the magnitude of $dR(\tilde{x}, \tilde{t})/d\tilde{t}$ decreases as time, $\tilde{t}$, increases. This indicates the system
can evolve towards equilibrium by balancing the effects of the source term with the decay caused by advection and diffusion. Notably, the system reaches equilibrium more rapidly when the diffusion timescale is shorter (i.e., $\left(\tilde{T}_{diff,x}^{-1} + \mu_i^2 T_{sys}\right)$ is larger).

To gain a more comprehensive understanding of the system, we conducted additional numerical experiments with the boundary conditions (32) and (33). The same parameter sets as in Figure 1 were used for these experiments. In Figure. 2., Panels (a)-(e) depict the time evolution of the precipitation field for three different parameter sets, while Panel (f) illustrates
the time evolution of spatially averaged fields, $\langle R(\tilde{x}, \tilde{t})\rangle$. Notably, we observed that the precipitation field with stronger advection effects (or weaker diffusion effects) converges more rapidly as shown in panel (f). In the case with strong advection effects with $u/V_{sys} = 0.5$ (black line in panel (f)), the decay due to the advection and the growth due to the source term becomes equalized, $d\langle R(\tilde{x}, \tilde{t})\rangle/d\tilde{t} \approx 0$, within $\tilde{t} \leq 1$. In the turbulent regime with $\tilde{L}_{diff,x}^{-1} \leq 1$, the growth due to the constant source term weakly depends on the strength of diffusion effects.






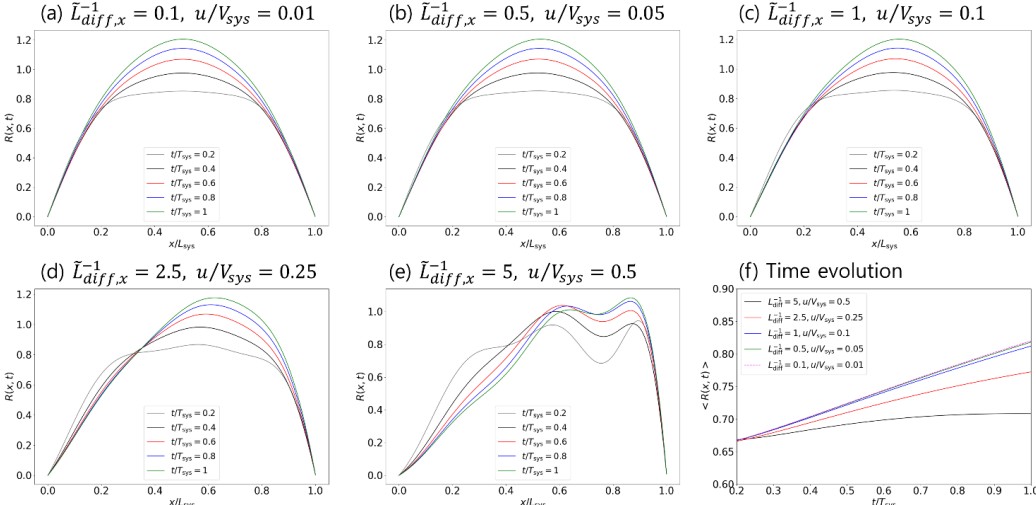

**Figure 2: The same as Figure 1 but including the stationary source-term.**

**4.4 Time-varying source-term and application for precipitation estimation**

The section introduces the application of solutions with arbitrary source terms, particularly focusing on atmospheric systems. In these systems, the development and decay of precipitation are often modeled using a fixed coefficient $\gamma$, expressed as $\sim\exp(-\gamma/T_{sys})$, to address the quasi-periodic nature of weather patterns (e.g., diurnal variations; Brutsaert 1974; Jinno et al. 1993). Under this assumption, the analytical solution maintains continuity throughout the entire domain, consistent with fluid properties. The significance of growth and decay within the diurnal cycle is emphasized through an error analysis of

precipitation nowcasting using MAPLE (Atencia et al. 2017). This effect is attributed to the solar cycle, with heating energy resulting in an increase in average rainfall in the afternoon. However, in atmospheric systems, there are instances of rapid field intensity development due to source terms within very short time scales ($t_s \ll T_{sys}$). Such source terms could significantly enhance the growth of precipitation, particularly in localized areas. Additionally, the regulation effects due to advection-diffusion on the field growth by the source term could be substantial for estimating the amount of precipitation more precisely.

In this context, the behavior of the analytic solutions with time-varying source terms is examined.

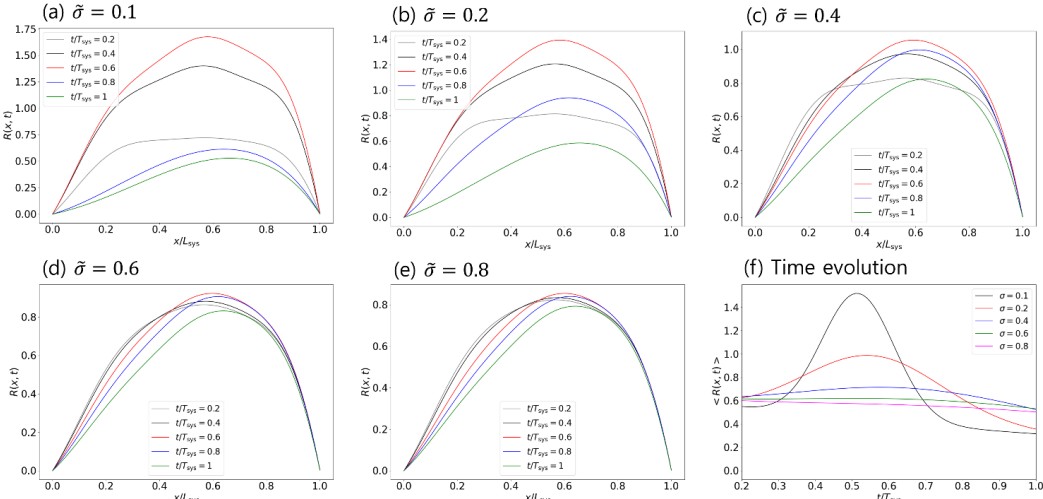

**Figure 3: Time evolution of field, $R(\tilde{x}, \tilde{t})$, including the time-varying source-term, $S(\tilde{x}, \tilde{t}) \approx \exp\left(-(\tilde{t} - \tilde{t}_s)^2/2\tilde{\sigma}^2\right)/\sqrt{2\pi\tilde{\sigma}^2}$ with $\tilde{\sigma}$, ranging from 0.1 to 0.8. Here, the parameter set ($\tilde{L}_{diff,x}^{-1} = 2.5$, $u/V_{sys} = 0.25$) for intermediate regime is used as an example.**

We first consider a straightforward case. When the contribution timescale of the source term, $t_s$, is significantly shorter than the system's dynamical scale, denoted as $\tilde{t}_s = t_s/T_{sys} \ll 1$, the source term can be simplified as $S(\tilde{x}, \tilde{t}) \approx N_s \delta(\tilde{t} - \tilde{t}_s)$ with the normalization constant, $N_s$. In this scenario, the contribution of the source term can be calculated as follows:

$$\bar{S}_i(\tilde{\tau}) \propto \exp\left(\tilde{T}_{diff,x}^{-1}\tilde{\tau}\right)\delta(\tilde{\tau} - \tilde{t}_s), (44)$$

$$I_{s,i}(\tilde{t}) \propto \int_0^{\tilde{t}} \delta(\tilde{\tau} - \tilde{t}_s)\exp\left(\left(\tilde{T}_{diff,x}^{-1} + \mu_i^2 T_{sys}\right)\tilde{\tau}\right)d\tilde{\tau}$$


$$= \exp\left(\left(\tilde{T}_{diff,x}^{-1} + \mu_i^2 T_{sys}\right)\tilde{t}_s\right)(2\Theta(\tilde{t}) - 1)\Theta(\tilde{t}_s - \tilde{t}\Theta(-\tilde{t}))\Theta(\tilde{t}\Theta(\tilde{t}) - \tilde{t}_s)$$

$$= \exp\left(\left(\tilde{T}_{diff,x}^{-1} + \mu_i^2 T_{sys}\right)\tilde{t}_s\right)\Theta(\tilde{t} - \tilde{t}_s), (45)$$

where, $\Theta(x)$ is the Heaviside step function,

$$\Theta(x) = \begin{cases} 0, & x < 0, \\ 1, & x \geq 0. \end{cases} (46)$$

Using Equation (45), we obtain the field,


$$R(\tilde{x}, \tilde{t}) \propto \sum_{i=1}^{\infty} \exp\left(-\left(\tilde{T}_{diff,x}^{-1} + \mu_i^2 T_{sys}\right)\tilde{t}\right)\left[\bar{\theta}_i(\tilde{t} = 0) + I_{s,i}(\tilde{t})\right]$$

$$= \sum_{i=1}^{\infty}\left\{\bar{\theta}_i(\tilde{t} = 0)\exp\left(-\left(\tilde{T}_{diff,x}^{-1} + \mu_i^2 T_{sys}\right)\tilde{t}\right)\right.$$

$$\left. + N_s\Theta(\tilde{t} - \tilde{t}_s)\exp\left(-\left(\tilde{T}_{diff,x}^{-1} + \mu_i^2 T_{sys}\right)(\tilde{t} - \tilde{t}_s)\right)\right\}. (47)$$

In this case, rapid growth only occurs at $\tilde{t} = \tilde{t}_s$, while the field intensity typically decreases throughout the spatial domain, where $0 \leq \tilde{t} \leq 1$.

To extend the analysis conducted earlier, we also consider a source term expressed as $S(\tilde{x}, \tilde{t}) \approx N_s \exp\left(-(\tilde{t} - \tilde{t}_s)^2/2\tilde{\sigma}^2\right)/\sqrt{2\pi\tilde{\sigma}^2}$, where $\tilde{t}_s$ represents the characteristic timescale of the source term, $\sigma$ denotes the standard deviation, and $N_s$ is the normalization constant. Here, we examine the effects of the source term on the system evolved by





advection and diffusion, focusing on the significance of source-term timescales in the growth of precipitation intensity. The contributions of the source term and the resulting field can be summarized as follows:

$$\bar{S}_i(\tilde{\tau}) \propto \frac{1}{\tilde{\sigma}} \exp\left(\tilde{T}_{diff,x}^{-1}\tilde{\tau} - \frac{(\tilde{\tau} - \tilde{t}_s)^2}{2\tilde{\sigma}^2}\right), (48)$$

$$I_{s,i}(\tilde{t}) \propto \int_0^{\tilde{t}} \frac{1}{\tilde{\sigma}} \exp\left((\tilde{T}_{diff,x}^{-1} + \mu_i^2 T_{sys})\tilde{\tau} - \frac{(\tilde{\tau} - \tilde{t}_s)^2}{2\tilde{\sigma}^2}\right) d\tilde{\tau}$$

$$\approx \exp\left(\frac{1}{2}(\tilde{T}_{diff,x}^{-1} + \mu_i^2 T_{sys})^2 \tilde{\sigma}^2 + (\tilde{T}_{diff,x}^{-1} + \mu_i^2 T_{sys})\tilde{t}_s\right)$$

$$\times \left[1.25331\left(\text{erf}\left(\frac{-0.707107(\tilde{T}_{diff,x}^{-1} + \mu_i^2 T_{sys})\tilde{\sigma}^2 + 0.707107(\tilde{t} - \tilde{t}_s)}{\tilde{\sigma}}\right)\right.\right.$$

$$\left.\left. - \text{erf}\left(\frac{-0.707107(\tilde{T}_{diff,x}^{-1} + \mu_i^2 T_{sys})\tilde{\sigma}^2 - 0.707107\tilde{t}_s}{\tilde{\sigma}}\right)\right)\right], (49)$$


$$R(\tilde{x}, \tilde{t}) \propto \sum_{i=1}^{\infty} \exp\left(-(\tilde{T}_{diff,x}^{-1} + \mu_i^2 T_{sys})\tilde{t}\right)\left[\bar{\theta}_i(\tilde{t} = 0) + I_{s,i}(\tilde{t})\right]$$

$$\approx \sum_{i=1}^{\infty} \left\{\bar{\theta}_i(\tilde{t} = 0) \exp\left(-(\tilde{T}_{diff,x}^{-1} + \mu_i^2 T_{sys})\tilde{t}\right)\right.$$

$$+ N_s \exp\left(\frac{1}{2}(\tilde{T}_{diff,x}^{-1} + \mu_i^2 T_{sys})^2 \tilde{\sigma}^2 - (\tilde{T}_{diff,x}^{-1} + \mu_i^2 T_{sys})(\tilde{t} - \tilde{t}_s)\right)$$

$$\times \left[1.25331\left(\text{erf}\left(\frac{-0.707107(\tilde{T}_{diff,x}^{-1} + \mu_i^2 T_{sys})\tilde{\sigma}^2 + 0.707107(\tilde{t} - \tilde{t}_s)}{\tilde{\sigma}}\right)\right.\right.$$

$$\left.\left.\left. - \text{erf}\left(\frac{-0.707107(\tilde{T}_{diff,x}^{-1} + \mu_i^2 T_{sys})\tilde{\sigma}^2 - 0.707107\tilde{t}_s}{\tilde{\sigma}}\right)\right)\right]\right\}. (50)$$

In Equation (50), the growth of the field due to the source term mainly depends on the term $\exp\left(\frac{1}{2}(\tilde{T}_{diff,x}^{-1} + \mu_i^2 T_{sys})^2 \tilde{\sigma}^2 - (\tilde{T}_{diff,x}^{-1} + \mu_i^2 T_{sys})(\tilde{t} - \tilde{t}_s)\right)$.

The behavior of numerical solutions with different forms of the source term, $S(\tilde{x}, \tilde{t})$ is shown in Figure 3. The boundary conditions, (32) and (33), and the parameter, $\tilde{\sigma}$, ranging from 0.1 to 0.4 with $\tilde{t}_s = 0.5$ were used. Note that $N_s$ is determined
by satisfying $\bar{\theta}_i(\tilde{t} = 0) \approx I_{s,i}(\tilde{t} = 0)$. For larger $\tilde{\sigma}$, the advection-diffusion effects become substantial on the evolution of $R(\tilde{x}, \tilde{t})$. The difference in $\tilde{\sigma}$ values can be interpreted as representing different types of precipitation. For instance, heavy convective rainfalls exhibited in a localized area could be modeled by the source term with a smaller $\tilde{\sigma}$, whereas stratiform rainfalls exhibited over a larger area could be modeled by the source term with a larger $\tilde{\sigma}$.



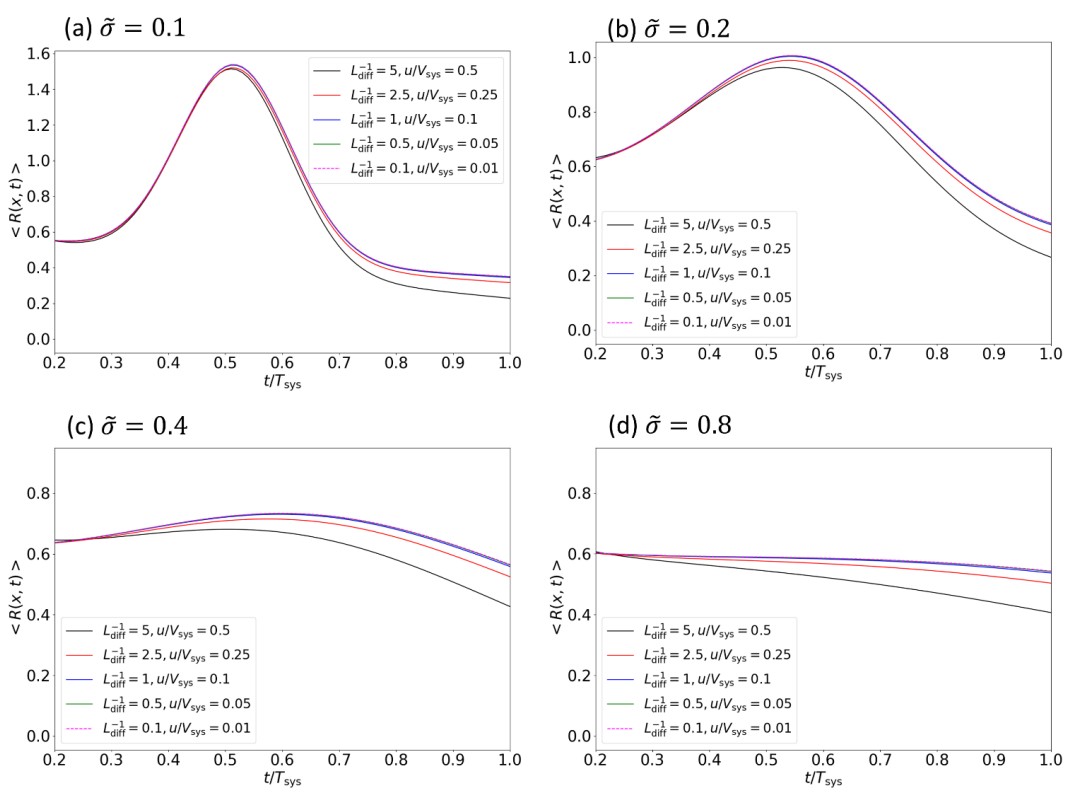


**Figure 4:** Time evolution of spatially averaged field, $\langle R(\tilde{x}, \tilde{t}) \rangle$, with different values of $\tilde{L}^{-1}_{diff,x}$, $u/V_{sys}$ and $\tilde{\sigma}$.

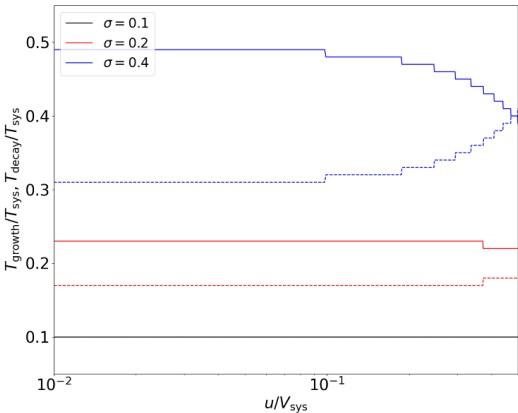

**Figure 5:** The timescales of growth (solid lines; $\tilde{T}_{growth}$) and decay (dashed lines; $\tilde{T}_{decay}$) as a function of advection speed, $u/V_{sys}$.

Note that $\tilde{T}_{growth} \approx \tilde{T}_{decay}$ in the case with $\tilde{\sigma} = 0.1$.

Figure 4 shows the time evolution of the spatially averaged field, $\langle R(\tilde{x}, \tilde{t}) \rangle$, with different values of $\tilde{\sigma}$, ranging from 0.1 to 0.8. The parameters $\left( \tilde{L}^{-1}_{diff,x}, u/V_{sys} \right)$ used to capture various effects of advection and diffusion are



$(0.1, 0.01), (0.5, 0.05), (1, 0.1), (2.5, 0.25),$ and $(5, 0.5)$. In cases with smaller $\tilde{\sigma}$ (panels (a) and (b)), the growth process

during $0 \leq \tilde{t} \leq \tilde{t}_s$ weakly depends on the advection effect, whereas the decay process is strongly affected by the advection effect. In the intermediate regime, the field decay due to advection and diffusion becomes dominant over the growth through the source term when $\tilde{\sigma}$ becomes larger (panel (d)). This is because for a sufficiently longer timescale, $\tilde{\sigma} \to \infty$, the source term can be approximated as $S(\tilde{x}, \tilde{t}) \to 0$. In the turbulent regime, on the other hand, the field evolution is consistent with the persistent evolution in Section 4.3.

In Figure 5, the growth and decay timescales as a function of the advection speed, $u/V_{sys}$, are shown with three different $\tilde{\sigma}$ values. Here, in the time domain including the source term effects, $\tilde{t}_s - \tilde{\sigma} \leq \tilde{t} \leq \tilde{t}_s + \tilde{\sigma}$, the timescales of growth and decay phases are defined as follows: (1) the timescale of growth phase: $\tilde{T}_{growth} = T_{growth}/T_{sys} = \tilde{t}_{max} - (\tilde{t}_s - \tilde{\sigma})$; (2) the timescale of decay phase: $\tilde{T}_{decay} = T_{decay}/T_{sys} = (\tilde{t}_s + \tilde{\sigma}) - \tilde{t}_{max}$. Here, $\tilde{t}_{max}$ indicates the timestep for maximum field intensity. The field is growing during $\tilde{t}_s - \tilde{\sigma} \leq \tilde{t} \leq \tilde{t}_s + \tilde{\sigma}$ when $\tilde{T}_{growth} > \tilde{T}_{decay}$. While the growth and decay timescales are almost comparable when $\tilde{\sigma}$ is very small (i.e., the case with $\tilde{\sigma} = 0.1$), both $\tilde{T}_{growth}$ and $\tilde{T}_{decay}$ depend on the advection and diffusion effects when $\tilde{\sigma}$ becomes larger. Notably, the decay timescale can be comparable to the growth timescale when $u/V_{sys}$ is sufficiently large, regardless of the presence of source term.

We further elaborate on the implications of these results for measuring quantitative precipitation estimation. The time-varying source term can be applied to quantitative precipitation estimation by obtaining $\tilde{\sigma}$ based on physical data. For example, deep convective clouds containing heavy precipitation can be detected using brightness temperature data from geostationary satellites (e.g., Kurino 1997; So & Shin 2018). Generally, larger convective clouds may lead to longer-lasting precipitation. Larger convective clouds often indicate stronger convective activity, resulting in the upward movement and cooling of moisture in the atmosphere. These conditions support the formation and sustenance of precipitation, potentially leading to a more prolonged period of precipitation. Additionally, a substantial fraction of growth and decay is attributed to orographic forcing (Foresti et al. 2018). By utilizing the size of deep convective clouds measured by geostationary satellites and the orographic information of the forecasting area, it is possible to empirically measure the corresponding $\tilde{\sigma}$. The source term derived from such $\tilde{\sigma}$ could be applicable for quantitative precipitation prediction.

### 4.5 Statistical treatment

In the realistic system, the physical ingredients for the source term could follow a distribution, and thus, a statistically averaged source term needs to be defined. This section describes the mathematical framework for calculating the statistically averaged source term and its influence on the system's response over time. Given the distribution of $\tilde{\sigma}$ in the domain $\tilde{\sigma}_1 \leq \tilde{\sigma} \leq \tilde{\sigma}_2$, denoted as $N(\tilde{\sigma})$, the statistically averaged source term can be defined as follows:

$$\langle S(\tilde{x}, \tilde{t}) \rangle \equiv \frac{\int_{\tilde{\sigma}_1}^{\tilde{\sigma}_2} S(\tilde{x}, \tilde{t}) N(\tilde{\sigma}) d\tilde{\sigma}}{\int_{\tilde{\sigma}_1}^{\tilde{\sigma}_2} N(\tilde{\sigma}) d\tilde{\sigma}}. \quad (51)$$

The statistically averaged contribution of the source term can be calculated through the equation (17):

$$\langle \bar{S}_i(\tilde{\tau}) \rangle = \int_0^1 \Psi_i(\tilde{x}) \langle S(\tilde{x}, \tilde{t}) \rangle \exp\left(-\bar{L}_{diff,x}^{-1} \tilde{x} + \tilde{T}_{diff,x}^{-1} \tilde{t}\right) d\tilde{x}, \quad (52)$$

$$\langle I_{s,i}(\tilde{t}) \rangle = \int_0^{\tilde{t}} \langle \bar{S}_i(\tilde{\tau}) \rangle \exp\left(\mu_i^2 T_{sys} \tilde{\tau}\right) d\tilde{\tau}. \quad (53)$$

$\langle \bar{S}_i(\tilde{\tau}) \rangle$ represents the statistically averaged contribution of the source term associated with the $i$-th component. It involves the integral of $\Psi_i(\tilde{x})$, the spatial profile of the component, multiplied by the statistically averaged source term and a decaying exponential term. $\langle I_{s,i}(\tilde{t}) \rangle$ indicates the integral of the statistically averaged contribution of the source term over time,





incorporating a time-dependent exponential factor. Using the equations (52) and (53), the field evolution can be described as follows:

$$R(\tilde{x}, \tilde{t}) \propto \sum_{i=1}^{\infty} \exp\left(-\left(\tilde{T}_{diff,x}^{-1} + \mu_i^2 T_{sys}\right)\tilde{t}\right)\left[\bar{\theta}_i(\tilde{t}=0) + \langle I_{s,i}(\tilde{t})\rangle\right]. (54)$$

**5  Summary and discussion**

This study concentrates on the application of the Advection-Diffusion equation, specifically analyzing fluid systems with time-varying source terms that drive rapid growth in field intensity. Utilizing analytic solutions obtained through integral transform techniques, we examined fluid systems within turbulent and intermediate regimes. Assuming that time-varying source terms follow a Gaussian distribution in the temporal domain, we investigated their dependence on source term characteristics, advection, and diffusion effects. In the turbulent regime with a sufficiently larger diffusion length, precipitation evolution weakly depends on source term characteristics and advection effects. In the intermediate regime with a relatively smaller diffusion length, advection becomes significant in the precipitation evolution. Particularly, it enhances the decay rate and regulates growth mediated by source terms. Diffusion also plays a crucial role in regulating the decay of the precipitation field.

While this study treated advection velocity as a stationary parameter to emphasize the relative importance of advection, diffusion, and source terms, a non-stationary velocity field could affect the relative importance of such effects. Here, we provide an intuition regarding the system evolution due to the presence of velocity perturbation. Assuming the velocity perturbation satisfying $\delta u \ll u$, we examined the characteristics of diffusion effects modified by the velocity perturbation:

$$\tilde{L}_{diff,x} = \frac{2D_x}{uL_{sys}}\left(1 + \frac{\delta u}{u}\right)^{-1} \approx \frac{2D_x}{uL_{sys}}\left(1 - \frac{\delta u}{u}\right), (55)$$

$$\tilde{T}_{diff,x} = \frac{4D_x}{u^2 T_{sys}}\left(1 + \frac{\delta u}{u}\right)^{-2} \approx \frac{4D_x}{u^2 T_{sys}}\left(1 - 2\frac{\delta u}{u}\right). (56)$$

Those equations represent that the positive perturbation, $\delta u > 0$, reduces the diffusion effects, whereas the negative perturbation, $\delta u < 0$, enhances the diffusion effects. Indeed, Ryu et al. (2020) argued that precipitation nowcasting using a non-stationary velocity field can enhance the accuracy of quantitative precipitation estimation, as demonstrated in a case study in the Korean Peninsula. The analysis presented in this paper could be extended by solving the coupled partial differential equations comprising Advection-Diffusion and Burgers' equations, which we leave for future work.

**Code availability**

The codes used for drawing the figures in manuscript are available upon the request to the corresponding author.

**Author contribution**

**Ji-Hoon Ha**: Conceptualization, Methodology, Investigation, Visualization, Writing – original draft, Writing – review & editing.





**Competing interests**

The authors declare that they have no conflict of interest.


**Acknowledgments**

This work was funded by the KMA Research and Development program "Developing AI technology for weather forecasting" under Grant (KMA 2021-00121).

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
