# Peer review of "Application of Advection-Diffusion Equation for Nonlinearly Evolving Precipitation Field"

_Nonlinear Processes in Geophysics, 2023_

## Referee Comment (RC2)

**A review on "Application of Advection-Diffusion Equation for Nonlinearly Evolving Precipitation Field" by Ji-Hoon Ha**
**(submitted to publication by Nonlinear Processes in Geophysics (NPG))**

Nonlinear dynamics of precipitation remains a challenging problem in nonlinear geophysics and it is therefore of great interest to NPG readership. Unfortunately, there is a significant gap between the title and the content of this paper.

This is already apparent in the literature review (section1), which is limited to rather operational techniques and does not address the current scientific deadlocks. Moreover, the assumed behaviour of the system (section 2) is merely categorised using global characteristic quantities (Eqs 4-6), which lead to the three classical regimes (turbulence, advection and diffusion dominant), while multiscale variability is not considered. The latter is fundamental for turbulence, which is rather dominant for atmospheric precipitation. This feature is still ignored when introducing the so-called analytic solution (section3). Indeed, the corresponding change of variables (Eq. 7) implicitly posits a very smooth behaviour that is highly debatable. This is further brought into question by the fixing of the parameters using the aforementioned global quantities (Eq.9) in the name of the "physical context", a claim that does not seem to be so obvious to me!

The potential interest of the method is illustrated by a series of case studies (section 4) ranging from a purely academic case (1D, no source) to slightly less academic cases (a stochastic, yet Gaussian source), but all with purely stationary advection. Unfortunately, no non trivial behaviour is highlighted in these examples, whose evolution graphs are, without any surprise, extremely smooth and unrealistically so.

Overall, I fear that this paper was not designed for NPG and that considerable revision work is needed to tailor it. This would require highlighting the limitations of the followed approach and, if possible, non trivial features revealed despite these limitations.